# In Vitro Assessment of Antiproliferative Activity and Cytotoxicity Modulation of *Capsicum chinense* By-Product Extracts

Lilian Dolores Chel-Guerrero [1,*], Matteo Scampicchio [2], Giovanna Ferrentino [2], Ingrid Mayanín Rodríguez-Buenfil [1,*] and Mabel Fragoso-Serrano [3,*]

[1] Centro de Investigación y Asistencia en Tecnología y Diseño del Estado de Jalisco A.C. Subsede Sureste, Tablaje Catastral 31264 Km, 5.5 Carretera Sierra Papacal-Chuburna Puerto Parque Científico Tecnológico de Yucatan, Merida 97302, Mexico

[2] Faculty of Science and Technology, Free University of Bozen-Bolzano, Piazza Università 5, 39100 Bolzano, Italy; matteo.scampicchio@unibz.it (M.S.); giovanna.ferrentino@unibz.it (G.F.)

[3] Departamento de Farmacia, Facultad de Química, Universidad Nacional Autónoma de Mexico (UNAM), Ciudad Universitaria, Ciudad de Mexico 04510, Mexico

[*] Correspondence: lchelg_al@ciatej.edu.mx (L.D.C.-G.); irodriguez@ciatej.mx (I.M.R.-B.); mabelfragoso@unam.mx (M.F.-S.)

**Abstract:** *Capsicum chinense* J., also known locally as habanero pepper, is a medicinal herb known for its pharmacological properties. Its properties are attributed to the capsaicinoids and polyphenols found in its fruit and polyphenols in its by-products. The anticancer potential of *C. chinense* by-products remains unexplored. This study aimed to evaluate the antiproliferative activity and modulation of the cytotoxicity of extracts obtained from *C. chinense* by-products of plants grown on black and red soils of Yucatan, Mexico. Dry by-product extracts were obtained using maceration, a Soxhlet, and supercritical fluid extraction. In vitro antiproliferative activity and cytotoxicity modulation were evaluated by the sulforhodamine B method. The extract of leaves of plants grown on black soil obtained by maceration displayed selective high cytotoxicity against colorectal cancer cells, $IC_{50}$ HCT–15 = 16.23 ± 2.89 μg mL$^{-1}$. The leaf and stem extracts of plants grown on red soil obtained by maceration potentiated the vinblastine's effect against parental breast cancer cells, MCF–7/Sens, with a reversion factor of 362.50-fold. Additionally, the extract of stems from plants grown on black soil obtained by supercritical fluid extraction and all the by-product extracts from plants grown on black soil obtained through maceration increased the effect of vinblastine against MCF–7/Vin$^+$ with a reversion factor from 5.06- to 7.78-fold. These results highlight the anticancer potential of *C. chinense* by-products.

**Keywords:** habanero; by-products; cytotoxicity; modulation; sensible and resistant MCF–7; supercritical fluid extraction; maceration; Soxhlet; soils

## 1. Introduction

In cancer, there is an abnormal proliferation of some cells. It causes nearly one on six deaths worldwide as the principal cause of mortality in advanced cancer patients is multidrug resistance. Moreover, cancer treatments can cause side effects, such as anemia, appetite loss, diarrhea, fatigue, nausea, vomiting, and general pain [1–4]. Thus, research is focusing on more effective treatments that can contribute to decrease the mortality rate.

Natural products (NPs) from medicinal plants are of great interest in drug discovery due to their specialized structures and their specific functions. They have been developed through natural evolution and are able to provide unusual features compared to conventional synthetic molecules [5]. NPs such as polyphenols, terpenoids, and coumarins can function like: chemopreventive drugs, chemotherapeutic drugs, sensitizers, and in reversing chemoresistance [6–12]. Among the several advantages of drugs derived from NPs, it is possible to include their availability, low cost, and effectiveness [13].

The diverse plants' genotypes synthesize natural products differentiated by type or quantity. The growing stage, environmental conditions, predation, and diseases are some factors that influence the biosynthesis of secondary metabolites. So, each plant, including each organ of a plant, could be a vast source of molecules for the development of chemotherapy drugs [14]. Nevertheless, there are many unexplored or few explored medicinal plants. One of these is *Capsicum chinense* Jacq variety Jaguar (commonly known as habanero pepper). This Solanaceae plant is native to the Americas. It was used in traditional medicine by the Aztecs and Mayans [15,16].

Nowadays, *C. chinense* is the main horticultural species commercially exploited in southeastern Mexico, with 358.37 ha cultivated (5049 T) [17] and has a designation of origin in the Yucatan Peninsula according to the Mexican Institute of Industrial Property. These plants are grown in two typical types of soils of Yucatan, "*K'áankab lu'um*" or red soils, and "*Box lu'um*", or black soils, differentiated by their organic and inorganic composition. The black soil has the highest content of calcium carbonates, organic matter, nitrogen, and phosphorus [18].

Yearly, around 7.9 million *C. chinense* plants and 155.3 million peduncles are discarded [17,19,20]. However, their leaves and stems are a rich source of bioactive compounds such as polyphenols, terpenoids, and coumarins. Moreover, their extracts have shown pharmacological properties such as antioxidant and anti-inflammatory activities [17,21,22]. Nevertheless, there is no research about the anticancer potential of these by-products. In this sense, it is important to evaluate the efficacy of different types of *C. chinense* by-product extracts as the bioactive compounds' content and the pharmacological activities are highly dependent on the type of extraction and soil where the plants were grown [18,22,23].

Currently, there is an interest in evaluating the bioactivity of extracts obtained using green technologies in comparison with the one obtained with conventional solvent extraction. Green technologies include supercritical fluid extraction, and conventional technologies include maceration [24–26].

Based on this background, the aim of the present investigation was to evaluate the antiproliferative activity and modulation effect of the cytotoxicity of extracts obtained from *Capsicum chinense* by-products of plants grown on black and red soils of Yucatan, Mexico. The by-products were leaves, stems, and peduncles of *C. chinense* variety Jaguar (Figure 1). Maceration, Soxhlet, and supercritical fluid were the methods used to obtain the extracts [17,22]. The extracts bioactivities against a panel of human cancer cell lines and a normal human breast cell line were evaluated.

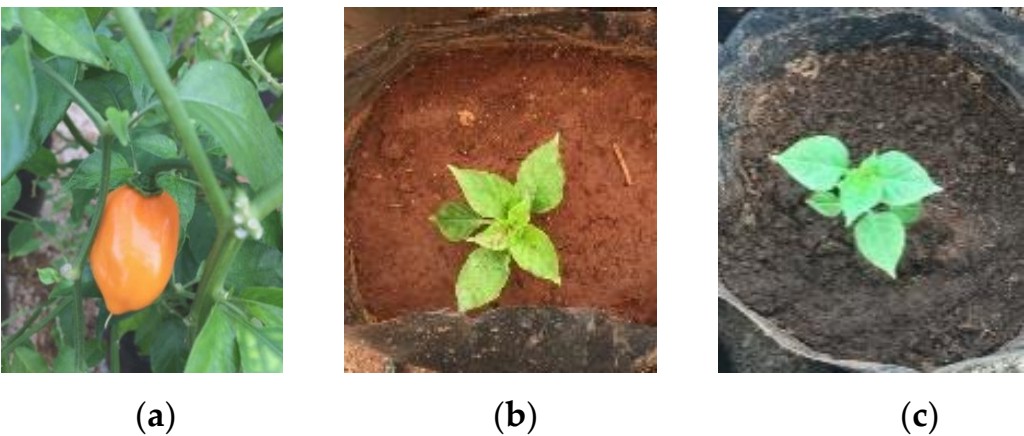

(a)        (b)        (c)

**Figure 1.** (**a**) *Capsicum chinense* (Jaguar variety), (**b**) *C. chinense* plant grown on red soil, and (**c**) *C. chinense* plant grown on black soil.

## 2. Materials and Methods

*2.1. Obtaining Extracts*

2.1.1. Plant Material

Thirty plants of *Capsicum chinense* J., variety Jaguar (variety register number CHL-008-101109) were cultivated in a greenhouse with a temperature from 24 to 47 °C and relative humidity of 91%, under controlled irrigation and fertilization conditions [17], using red soil (*K'áankab lu'um*) and black soil (*Box lu'um*), acquired from a supplier in Merida, Yucatan, Mexico. For that purpose, seedlings grown for 45 days with a minimum height of 19.3 cm and ten true leaves were used. They were obtained from the Cutz nursery in Suma de Hidalgo, Yucatan, Mexico. From there, they were transplanted in polyethylene bags filled with 12 kg of each type of soil (50% of the plants for each one). After the last expected harvest of fruits, 265 days of transplantation (DAT), leaves, stems, and peduncles were collected [18]. The greenhouse was located in the Centro de Investigación y Asistencia en Tecnología y Diseño del Estado de Jalisco, A.C. (CIATEJ) Subsede Sureste, Merida, Yucatan, Mexico (latitude N 21°8′1.288″ and longitude W 89°46′52.26″).

2.1.2. Drying of *Capsicum chinense* By-Products

From *Capsicum chinense* plants the peduncles, stems, and leaves were separated. Subsequently, peduncles and stems were dried at 44 °C for 48 h and leaves at 44 °C for 240 h in a stainless-steel oven (HS60-AID, Novatech, Jalisco, Mexico) to reach at least 5% of moisture. Then, the dried samples were ground in a blender, sieved (pore size 500 μm, Sieve # 35, Fisher Scientific, Boston, MA, USA) and stored at −20 °C until the day of the analysis according to the methodology reported by Chel-Guerrero et al. [22].

2.1.3. Maceration Extraction (ME)

For the extraction by maceration, HPLC-grade methanol (Sigma Aldrich, Naucalpan de Juarez, Mexico) was used. The procedure was performed according to the methodology reported in Chel-Guerrero et al. [25]. Briefly, 5 g of each sample was covered with 50 mL of the solvent and shaken for 24 h at 28 °C at 160 rpm in a shaking incubator (Labtech brand model LSI-3016A, Jalisco, Mexico). Subsequently, the samples were filtered with Whatman No. 2 paper. The solvent was eliminated with a rotary evaporator under vacuum at 40 °C (model B-491, Buchi brand, Flawil, Switzerland). The dried extracts were stored at –20 °C until their analysis.

2.1.4. Soxhlet Extraction (SOX)

The Soxhlet extraction was carried out according to the methodology described by Paes et al. [27]. Briefly, 5 g of each dried sample was loaded in a Soxhlet apparatus. Then, 150 mL of the solvent ethanol was recycled for three hours at 78 °C.

2.1.5. Supercritical Fluid Extraction (SFE)

The SFE extracts were obtained following the method described in the work of Chel-Guerrero et al. [17]. Briefly, 15 g of each dried by-product was extracted with $CO_2$ plus ethanol (5% *w/v*) with a flow rate equal to 2 L $h^{-1}$, at 30 MPa, 45 °C for two hours, using a supercritical fluid extraction system (Superfluidi s.r.l., Padova, Italy). The extracts were collected in a 250 mL volumetric flask and the ethanol was removed using a rotary evaporator.

*2.2. Cell Lines*

All drug-sensitive cell lines, including HeLa (ATCC CCL-2), MCF–7 (ATCC HTB-22), HCT–15 (ATCC CCL-225), HCT–116 (ATCC CCL-247), Caov3 (ATCC HTB-75) and 184B5 (ATCC CRL-8799), were acquired from the American Type Culture Collection (Manassas, VA). The resistant counterpart MCF–7/Vin was developed through continuous exposition to vinblastine for nine consecutive years.

### 2.3. Cytotoxic Activity

The cytotoxicity was evaluated using the sulforhodamine B (SRB) assay, testing against a panel of human cancer cell lines: cervical uterine (HeLa), breast (MCF–7), colorectal (HCT–116 and HCT–15), ovarian (Caov-3), and a normal breast epithelium 184B5 cells according to the methodology reported by Skehan et al. [28]. A density of $5 \times 10^3$ cells in 96-well plates in a volume of 190 μL was used. After, 10 μL of the test samples at 0.2, 1, 5, and 25 μg mL$^{-1}$ were added. The samples were incubated for 72 h at 37 °C in a humidified atmosphere of 5 % $CO_2$. Vinblastine was used as a positive control (0.0032, 0.016, 0.4, and 2 μg mL$^{-1}$).

### 2.4. Modulation of Cytotoxicity

The evaluation of the modulation of the cytotoxicity was performed using the SRB method [29] against sensitive or parental breast cancer cells (MCF–7), resistant breast cancer cells growing in absence of vinblastine (MCF–7/Vin$^-$), and resistant breast cancer cells growing in presence of 0.19 μg mL$^{-1}$ vinblastine (MCF–7/Vin$^+$). The half-maximal inhibitory concentration (IC$_{50}$) and the reversal fold value (the ratio of the IC$_{50}$ of vinblastine alone to the IC$_{50}$ of vinblastine with the tested extract) were determined as indicators of the extracts' capacity to improve the cytotoxicity of vinblastine.

A density of $5 \times 10^3$ cells in 96-well plates in a volume of 180 μL was used. After, 10 μL of each concentration of vinblastine (serial dilutions from 0.000128 to 2 μg mL$^{-1}$) and 10 μL of each sample at 25 μg mL$^{-1}$ or reserpine at 5 μg mL$^{-1}$ were added. The samples were incubated for 72 h at 37 °C in a humidified atmosphere of 5 % $CO_2$. Reserpine was used as a positive control (5 μg mL$^{-1}$).

### 2.5. Statistical Analysis

The experiments were carried out in triplicate. The results were expressed as mean values ± standard deviations. Statistically significant differences between groups were evaluated through the analysis of variance followed by Tukey's test ($p \leq 0.05$). For these analyses, Statgraphics Centurion 18 X64 software was used (Statgraphics software, The Plains, VA, USA). Nonlinear regression of individual experiments was determined to calculate the average values of IC$_{50}$. For this nonlinear regression, GraphPad Prism 9 software was used (GraphPad Software, La Jolla, CA, USA). For the principal components analysis with standardization of the variables [30], the Statgraphics 19 software (Statgraphics software, The Plains, VA, USA) was used. Pearson tests were carried out to determine correlations and their significance.

## 3. Results

### 3.1. Antiproliferative Activity

In the present study, the efficacy of the extracts of Habanero pepper using an SRB assay, a method applied by the National Cancer Institute's (USA-NCI) compound screening program, was evaluated [28]. According to the National Cancer Institute Plant Screening Program, a crude extract has in vitro cytotoxic activity if the IC$_{50}$ value is ≤20 μg mL$^{-1}$ after incubation for 48 and 72 h [31,32]. Based on this scale, all the extracts obtained in the present study were noncytotoxic against HeLa, MCF–7, HCT–116, HCT–15, and Caov-3 cancer cell lines, and the 184B5 breast normal cell line, except for the sample of leaves of *C. chinense* grown on black soil obtained by maceration with methanol (LBS ME), which exhibited cytotoxic activity against the HCT–15 cell line with IC$_{50}$ = 16.23 ± 2.89 μg mL$^{-1}$ (Figure 2). It was also categorized as highly active according to the criteria proposed by Srisawat et al. [33] by evaluating the cytotoxic activity of plant by-products against human breast cancer cell lines. Additionally, the results revealed that this extract possessed significant specificity against HCT–15 cells compared to the other cancer cells and no toxicity against normal cells.

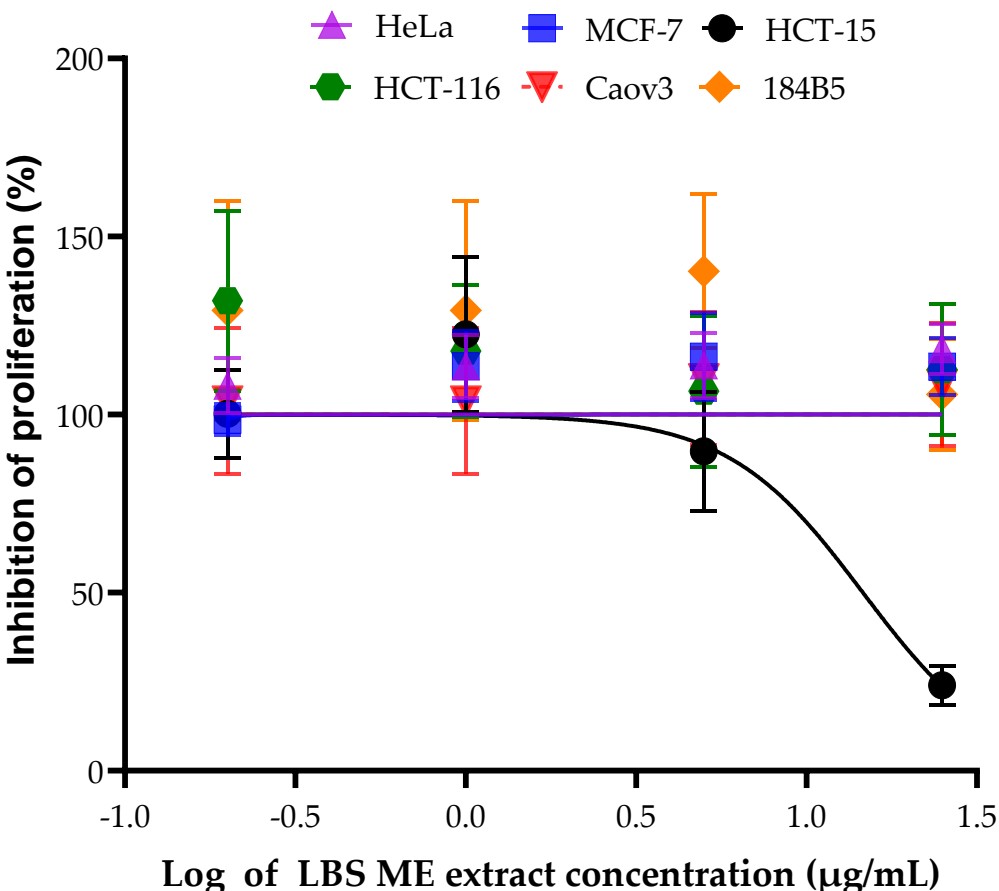

**Figure 2.** Dose–response of extracts of leaves of *C. chinense* grown on black soil (LBS), obtained by maceration (n = 3) against cancer cell lines (HeLa, MCF–7, HCT–15, HCT–116, Caov3) and breast normal cell line (184B5). $IC_{50}$ HCT–15 = 16.23 ± 2.89 μg mL$^{-1}$.

The results of this study were similar to those reported by Jeon et al. [34]. They mentioned that the methanol extract of leaves from *Capsicum annuum* L. exhibited antiproliferative activity, as determined by an MTT assay, against colorectal cancer cell line (HCT–116), with $IC_{50}$ values of 80, 38, and 23% and against breast cancer cell line (MCF–7) with $IC_{50}$ values of 78, 37, and 26% at the concentration of 1, 0.5, and 0.25 μg mL$^{-1}$, respectively. Furthermore, the results indicate that the observed bioactivity of each extract could be due to the presence of the different compounds contained in the plant materials as they belong to distinct species [35].

Moreover, according to Srisawat et al. [33], our results indicated that methanol was the best solvent for extraction, in terms of the cytotoxic properties of the extracts, probably due to the highly polar bioactive compounds extracted [12,21,36,37].

In particularly, in the samples studied, black soil, a loamy sand soil, unlike red ground, a clay loam soil with a medium texture, favored the presence of compounds with intermediated and high polarity, including polyphenols. This was probably associated to the high content of manganese, organic matter, and nitrogen present in the soil. These components have shown an increase in enzyme phenylalanine ammonia-lyase (PAL) activity, which has a significant role in the biosynthesis of polyphenols for plants of the genus *Capsicum* [38]. In this sense, the extract of leaves of *C. chinense* grown on black soil obtained by maceration with methanol contained vanillin, myricetin, rutin, kaempferol, quercetin + luteolin, hesperidin + diosmin and neohesperidin, as well as chlorogenic acid, coumaric acid, *p*-coumaric acid, and cinnamic acid. Their concentrations were 46.20, 461.47, 99.43, 204.07, 506.77, 122.57, 28.53, 308.53, 123.90, 94.17, 125.93 mg 100 g$^{-1}$ dry basis, respectively. They are compounds that have exhibited anticancer activity [17]. Nevertheless, the Pearson

correlation coefficient calculated for these polyphenol contents and the cytotoxicity of the samples evaluated in our research showed a very low correlation with values ranging between $-0.29$ and $0.21$, implying that the polyphenols were not responsible for the cytotoxic activity.

On the other hand, Chel-Guerrero et al. [17] indicated that the Soxhlet method extracted more polyphenols from *Capsicum chinense* by-products than the maceration method. Moreover, apigenin and diosmetin were extracted from these samples, compounds that have antiproliferative activity [10,39,40]. Despite this, only maceration extracts exhibited antiproliferative activity. So, this also confirmed that the polyphenols in these samples were not responsible for the reported antiproliferative activity. This could be due to the concentration of these compounds in the extracts or because other compounds masked the polyphenols action. Another cause could be the high temperature used in the Soxhlet method, which could cause the loss of thermolabile compounds. Moreover, the antiproliferative activity of LBS ME extracts against colorectal cells could be associated to other compounds, such as triterpenoids or coumarins [6,7,24,41–44] or to a synergism among all of them [6].

As concerns the other samples, although they also contained similar types of compounds, such as flavonoids, coumarins, and terpenoids and in particular similar polyphenols as for the LBS ME extract, they did not show any bioactivity. Many factors could explain this fact, such as the different quantities of compounds present in them. It may also be that the secondary metabolites responsible for the activity were absent or that certain compounds present in the samples could have masked the bioactivity of others. Another explanation is that the mixture of compounds producing a synergistic cytotoxic effect was different in each of the extracts analyzed [25,45].

Interestingly, none of the analyzed samples exhibited cytotoxic activity against the breast cell line (184B5) compared to the drug used as the positive control (vinblastine). Thus, future research activities are necessary to identify those compounds responsible for cytotoxicity and to confirm why such extracts did not show toxicity for humans.

These results suggested that compounds from the leaves of plants grown on black soil may serve as a promising new experimental anticancer agent. However, further research on in vivo models is necessary to confirm their anticancer activity and their mechanism of action.

### 3.2. Modulation of Cytotoxicity

The multidrug resistance (MDR) phenotype is considered a significant cause of failure in cancer treatment. MDR is usually mediated by the overexpression of drug efflux pumps of a P-glycoprotein. Compounds that mitigate the MDR phenotype by modulating the activity of these transport proteins are important targets [46]. We tested all samples as modulators of efflux pumps in vinblastine-resistant MCF–7/Vin$^+$ cells. At the same time, a stock of MCF–7/Vin$^-$ cells were maintained in a vinblastine-free medium. We calculated the reversion factor, a potency parameter, as the ratio between the $IC_{50}$ of vinblastine alone and the value of the $IC_{50}$ of vinblastine plus the tested compounds [47].

Table 1 exhibits the results of cytotoxic modulation against MCF–7/Sens, MCF–7/Vin$^-$ and MCF–7/Vin$^+$ cell lines.

The leaves and stems extracts of plants grown on red soil obtained by ME increased the effect of the drug vinblastine against parental breast cancer cells, MCF–7/Sens, with a reversion factor of 362.50-fold (Table 1 and Figure 3).

The extract obtained using SFE from the stems of plants grown on black soil and all by-product extracts from plants grown on black soil obtained by ME exhibited a strong modulatory effect of cytotoxic activity against MCF–7/Vin$^+$ cells, with an RF from 5.06- to 7.78-fold (Table 1 and Figure 4).

**Table 1.** Modulating effect of the cytotoxic activity of habanero by-product extracts on human cell lines of breast cancer, parental and resistant to vinblastine.

| Type of Habanero By-Product or Drug [A] | MCF–7/Sens [B] IC$_{50}$ (µg mL$^{-1}$) | RF [E] | MCF–7/Vin$^{-}$ [C] IC$_{50}$ (µg mL$^{-1}$) | RF | MCF–7/Vin$^{+}$ [D] IC$_{50}$ (µg mL$^{-1}$) | RF |
|---|---|---|---|---|---|---|
| PBS ME | 0.0022 ± 0.0006 | 32.95 [bcd] | 0.46 ± 0.02 | 3.80 [bc] | 0.32 ± 0.07 [g] | 5.77 [b] |
| LBS ME | 0.0470 ± 0.0023 | 4.93 [d] | 0.38 ± 0.01 | 4.49 [b] | 0.36 ± 0.02 [fg] | 5.06 [b] |
| SBS ME | 0.0014 ± 0.0003 | 51.79 [bc] | 2.01 ± 0.33 | 0.87 [gh] | 0.23 ± 0.04 [g] | 7.78 [a] |
| PRS ME | 0.0010 ± 0.00006 | 72.50 [b] | 0.56 ± 0.04 | 3.15 [bcd] | 0.95 ± 0.07 | 1.94 [cd] |
| LRS ME | 0.0002 ± 0.00002 | 362.50 [a] | 0.61 ± 0.02 | 2.85 [d] | 0.55 ± 0.05 | 3.33 [c] |
| SRS ME | 0.0002 ± 0.00004 | 362.50 [a] | 0.59 ± 0.02 | 2.93 [cd] | 0.89 ± 0.02 | 2.06 [cd] |
| LBS SFE | 0.0045 ± 0.0012 | 16.11 [cd] | 3.74 ± 0.29 | 0.47 [h] | 1.73 ± 0.21 | 1.07 [d] |
| SBS SFE | 0.2213 ± 0.0065 | 0.33 [d] | 0.43 ± 0.14 | 4.05 [b] | 0.28 ± 0.09 | 6.59 [ab] |
| SRS SFE | 0.0141 ± 0.0031 | 5.14 [d] | 0.72 ± 0.16 | 2.40 [dc] | 1.84 ± 0.23 | 1.00 [d] |
| LBS SOX | 0.0084 ± 0.0017 | 8.63 [cd] | 0.94 ± 0.05 | 1.85 [ef] | 1.48 ± 0.06 | 1.24 [d] |
| SBS SOX | 0.0069 ± 0.0010 | 10.51 [cd] | 1.12 ± 0.14 | 1.56 [fg] | 1.19 ± 0.35 | 1.54 [d] |
| SRS SOX | 0.0036 ± 0.0006 | 20.35 [cd] | 0.46 ± 0.10 | 3.75 [bcd] | 1.96 ± 0.07 | 0.94 [d] |
| Reserpine [F] | 0.0022 ± 0.0001 | 32.95 [bcd] | 0.078 ± 0.01 | 22.42 [a] | 0.58 ± 0.05 | 3.14 [c] |
| Vinblastine | 0.0725 ± 0.0028 | | 1.75 ± 0.03 | | 1.84 ± 0.02 | |

[A] Serial dilutions from 0.000128 to 2 µg mL$^{-1}$ of vinblastine in the presence or absence of extract (25 µg mL$^{-1}$); P = peduncles; L = leaves; S = stems; BS = black soil; RS = red soil; ME = extraction by maceration; SFE = supercritical fluid extraction; SOX = extraction by Soxhlet; [B] MCF–7/Sens = sensitive or parental breast cancer cells; [C] MCF–7/Vin$^{-}$ = resistant breast cancer cells that grow up in the absence of vinblastine; [D] MCF–7/Vin$^{+}$ = resistant breast cancer cells that grow up in the presence of 0.19 µg mL$^{-1}$ of vinblastine). [RE] F = reversal factor (IC$_{50}$ vinblastine/IC$_{50}$ vinblastine in the presence of extract). [F] Reserpine = 5 µg mL$^{-1}$ as positive control; each value represents the mean ± SD of three independent experiments. [a–h] Different superscript letters in the same row indicates statistically significant differences ($p \leq 0.05$).

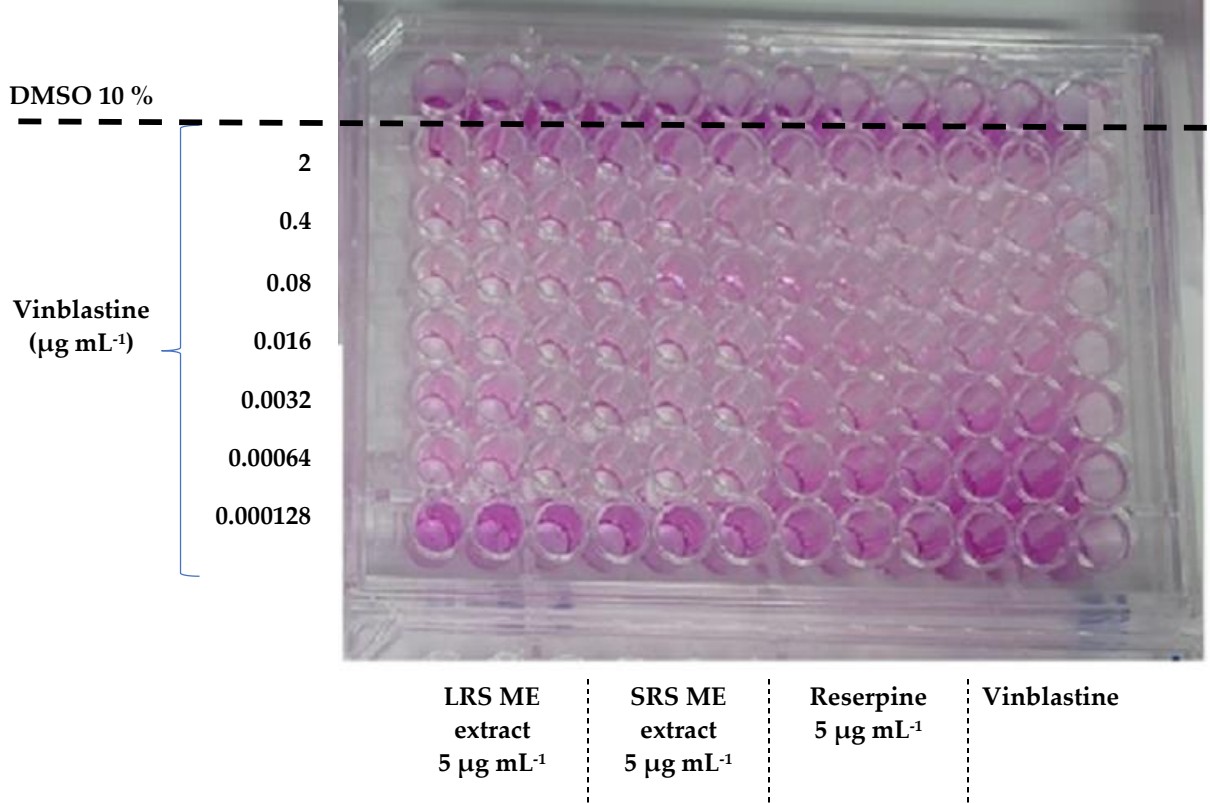

**Figure 3.** Modulation assay of vinblastine with leaves and stems of plants grown on red soil extracts obtained using maceration against MCF–7/Sens parental cell line. Extracts with the highest reversal factor against MCF–7/Sens (RF = 362.50).

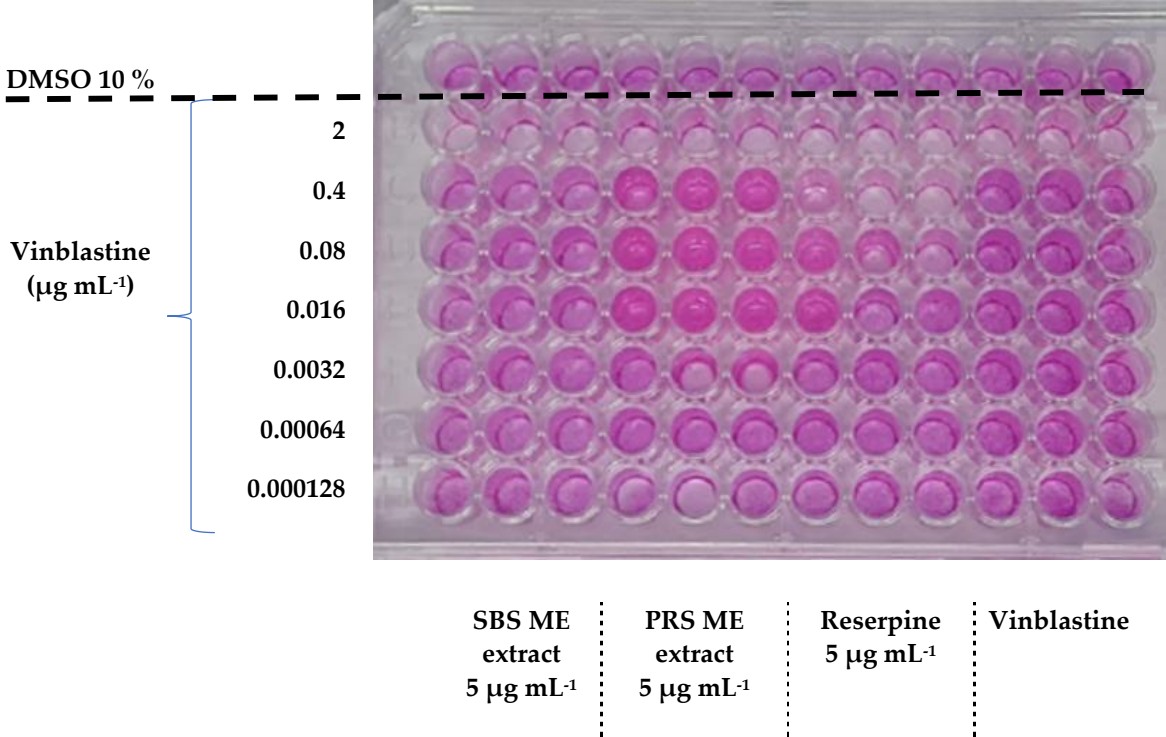

**Figure 4.** Modulation assay of vinblastine with stems of plants grown on black soil and peduncles of plants grown on red soil extracts obtained using maceration against MCF–7/Vin$^+$-resistant cell line. SBS ME: extract with the highest reversal factor against MCF–7/Vin$^+$ (RF = 7.78).

The result is similar or superior to the one observed in the reserpine, positive control (RF 32.95 for MCF–7/Sens and 3.14 for MCF–7/Vin$^+$), a cytotoxic positive efflux pump control [29,47,48]. Additionally, all the extracts showed a modulation effect against MCF–7/Vin$^-$ (RF from 0.47- to 4.05-fold), except the stem and leaf extracts of plants grown on black soil obtained using ME and SFE, respectively. These extracts had a lower RF score than the positive control (RF 22.42-fold).

These results were in line with those of Lin et al. [6]. They mentioned that herbal extracts such as *Solanum nigrum* and *Claviceps purpurea*, combined with chemotherapy drugs, attenuated the resistance to the drug and exerted chemoprotective actions. The reversal activity of our samples could be due to an additive synergism. Natural compounds applied with chemotherapy drugs increase the cytotoxic effects of known anticancer agents. They exert their functions in multiples ways, e.g., through autophagy induction, via regulating pro-inflammatory cytokines such as tumor necrosis factor α (TNF-α) and interferon γ (IFN-γ), by modulating the tumor micro-environment, or by regulating the expression or activity of the transcription nuclear factor kappa B (NF-κB). NF-κB seems to be an indicator for determining the potency of chemotherapeutic cytotoxicity [6,49]. In the extracts studied, further research is necessary to determine the mechanisms of action used by the compounds present in the samples to potentiate the vinblastine cytotoxicity.

In addition, the Pearson correlation coefficient calculated for the polyphenol content previously quantified by Chel-Guerrero et al. [17], and the cytotoxicity modulation of the samples evaluated in the present research, showed a very low correlation with values ranging between −0.51 and 0.06 for modulation against MCF–7/Sens and with values ranging between −0.88 and 0.16 for modulation against MCF–7/Vin$^-$. For modulation against MCF–7/Vin$^+$, we obtained values from −0.51 to 0.58, a moderately positive correlation between the cytotoxicity modulation of the samples and gallic acid and catechin content (0.48 and 0.49, respectively) and a strong positive correlation between modulation and protocatechuic acid (0.58). Hence, the modulation of the cytotoxicity against MCF–7/Sens

could be explained by other types of compounds present in the samples [22], such as terpenoids or coumarins [6,7]. Gallic acid, catechin, and protocatechuic acid could be responsible for the cytotoxicity modulation against MCF–7/Vin$^+$.

On the other hand, the methanol maceration method seems to subserve the extraction of compounds with anticancer potential from *Capsicum chinense* by-products. Thus, the compounds responsible for these biological activities are probably highly polar and thermolabile [36,37,50] For this reason, Soxhlet with ethanol and extraction by supercritical fluids with $CO_2$ + ethanol did not extract them. Additionally, there were significant differences due to the type of soil in which the plants were grown for extracts having antiproliferative activity and modulating effect on cytotoxic activity. Black soil used for *C. chinense* by-products showed cytotoxicity against HCT–15 cancer cells and cytotoxicity modulation on MCF–7/Vin$^+$, while the red soil showed to potentiate the cytotoxicity of vinblastine against MCF–7/Sens.

Moreover, a principal components analysis was carried out with standardization of the variables [30] to establish the amount of variance associated with the components integrated by MCF–7/Sens, MCF–7/Vin$^-$ and MCF–7/Vin$^+$ (Table 2).

**Table 2.** Principal component analysis.

| Number of Component | Eigenvalue [a] | Variance Percentage | Accumulated Percentage |
|---|---|---|---|
| 1 | 1.5 | 50.6 | 50.6 |
| 2 | 0.8 | 27.1 | 77.7 |
| 3 | 0.7 | 22.3 | 100.0 |

[a] The eigenvalues are proportional to the percentage of variability in the data attributable to the components.

According to the previous table, the first component explains 50.6% of the variance, and the second explains 27.1%. So, together, both approach the 80% acceptable for descriptive purposes.

Based on the eigenvectors (Table 3), on PC1, the influence of MCF–7/Vin$^+$ was the greatest in a positive direction. It was followed by MCF–7/Vin$^-$ in the same positive way. In the opposite direction was the impact on PC1 of MCF–7/Sens.

**Table 3.** Eigenvectors of the components from response variables.

| Variable | Component 1 | Component 2 | Component 3 |
|---|---|---|---|
| MCF–7/Sens | −0.575434 | 0.588499 | 0.567930 |
| MCF–7/Vin$^-$ | 0.531068 | 0.796973 | −0.287752 |
| MCF–7/Vin$^+$ | 0.621967 | −0.136027 | 0.771138 |

MCF–7/Vin$^-$ was the highest and followed a positive direction, followed by MCF–7/Sens in the same way. In the opposite direction, the impact of MCF–7/Vin$^+$ was the smallest (−0.136) and followed the opposite direction.

For the modulation of *C. chinense* by-products, this analysis allowed us to determine that was more relevant the result obtained for MCF–7/Vin$^-$ and MCF–7/Vin$^+$ than those obtained for MCF–7/Sens as MCF–7/Vin$^-$ and MCF–7/Vin$^+$ were the resistant cell lines.

## 4. Conclusions

This work showed the potential of leaves of *C. chinense*, a medicinal plant grown on black soil, as a cytotoxic agent specifically against human colorectal cancer cells. The results also demonstrated that the *C. chinense* by-products could aid the discovery of new MDR-modifying leads. The modulatory effects were like those displayed by the positive control, reserpine. However, it is necessary to pursue further pharmacological studies to identify and purify the active compounds for subsequent in vivo evaluation of their activity and the identification of their mechanisms of action.

**Author Contributions:** Conceptualization, Methodology, Software, Data curation, Investigation, Resources, Funding acquisition, Writing—reviewing and editing, L.D.C.-G., M.F.-S., M.S., G.F. and I.M.R.-B.; Formal analysis, Writing—original, Draft preparation, Visualization, Project administration, L.D.C.-G. and M.F.-S. All authors have read and agreed to the published version of the manuscript.

**Funding:** This research was funded by the Consejo Nacional de Ciencia y Tecnología de México (CONACyT), Centro de Investigaciones y Estudios Superiores en Antropología Social (CIESAS), and the International Development Research Centre of Canada (IDRC Canada), project No. 50330, and a research scholarship granted to Lilian Dolores Chel Guerrero, and by the Consejo Nacional de Ciencia y Tecnología de Mexico (CONACyT), project No. 257588.

**Institutional Review Board Statement:** Not applicable.

**Informed Consent Statement:** Not applicable.

**Data Availability Statement:** The data used to support the findings of this study are included within the article.

**Acknowledgments:** We acknowledge Rogelio Pereda-Miranda for permitting us to perform the bioactivity assays in his laboratory at the Faculty of Chemistry of the Universidad Nacional Autónoma de Mexico (DGAPA IN 208019), we thank Luis Antonio Chel-Guerrero of the Faculty of Chemical Engineering of the Universidad Autónoma de Yucatan for his support in the PCA analysis and we also thank Gabriela Castañeda-Corral from the Universidad Autónoma del Estado de Morelos for the orientation in the redaction of the manuscript.

**Conflicts of Interest:** The authors declare no conflict of interest. The funders had no role in the design of the study; in the collection, analysis, or interpretation of data; in the writing of the manuscript; or in the decision to publish the results.

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
