# Peer review of "In Vitro Assessment of Antiproliferative Activity and Cytotoxicity Modulation of Capsicum chinense By-Product Extracts"

_applsci, doi:10.3390/app12125818_

Round 1

Reviewer 1 Report

Journal: Applied Sciences

Manuscript ID: 1742597

In vitro study of the antiproliferative activity and modulation of cytotoxicity of C. chinense by-products extracts obtained from plants grown on black and red soils of Yucatán, México

Comments

In this work, the authors investigated the anticancer potentials of C. chinense by-product extracts of plants that were grown on black and red soil of a particular region in Mexico. The methods of maceration, Soxhlet and Supercritical Fluid Extraction (SFE) were used to obtained the extracts of plant. Sulforhodamine B method  is employed to evaluate the antiproliferative activity and modulation of cytotoxicity. The different extracts exhibited the significant anticancer potentials. These findings have medical significance as well are usful for public awareness. For the publication of the manuscript some technical corrections are required.

1.     Better to rewrite a short and concise title.

2.     The extracts obtain from plants grown in a different fields i.e. red and black solid demonstrated different activities. It is required to discuss with evidence the key factors that alter the activities of extracts such a texture, mineral contents, fertility, moisture , and nutrients availability and the type of soil etc focusing solely the test samples. Although several comments are included from literature for support.

3.     Why only the methanol extracts obtain from the leave of C. chinense grown in black soil obatianed by merceration exhibited cytotoxic activity against HCT-15 cells? While other extracts were found no-cytotoxic. Discuss the key factors that made this extract cytotoxic.

4.     Recheck the remove language error such as format, speeling, grammer mistakes etc. E.g line 150 capital C of Coumaric should be small c. At line 226 “by-product from” should be “by-product of”. At line 234 “modulation and” should be “modulation of”. Some sentences are not clear rewrite them in simple understable way such as sentence from lin 263 to 266.

5.     In the sentence between line 154-157 mentioned that cytotoxicity is not due to polyphenols. If then what are other natural products present playing the role of cytotoxicity? However, major polyphenols are cytotoxic. There is a contrast of comment on line 154-157, with line 164-166. Justify your comment.

6.     Why the reversion factor increased from 32.95 to 362.50 fold? Seems very large change.

7.     Better to add assay images of methanol extract to demonstrate potency.

8.     I recommend the GC-MS analysis of most potent methanol extract for the identification of bio-active components.

 A major revision is required following the above comments to meet the quality of publication.

Author Response

  1. Better to rewrite a short and concise title.

Authors’ response: We write a short title

  1. The extracts obtain from plants grown in a different fields i.e. red and black solid demonstrated different activities. It is required to discuss with evidence the key factors that alter the activities of extracts such a texture, mineral contents, fertility, moisture , and nutrients availability and the type of soil etc focusing solely the test samples. Although several comments are included from literature for support.

Authors’ response:  We include more information about key factors related with soild based on literature, because it was not the objective of our study.

  1. Why only the methanol extracts obtain from the leave of  chinensegrown in black soil obatianed by merceration exhibited cytotoxic activity against HCT-15 cells? While other extracts were found no-cytotoxic. Discuss the key factors that made this extract cytotoxic.

Authors’ response: We include this in the discussion

  1. Recheck the remove language error such as format, speeling, grammer mistakes etc. E.g line 150 capital C of Coumaric should be small c. At line 226 “by-product from” should be “by-product of”. At line 234 “modulation and” should be “modulation of”. Some sentences are not clear rewrite them in simple understable way such as sentence from lin 263 to 266.

Authors’ response: We recheck language error and we correct them.

  1. In the sentence between line 154-157 mentioned that cytotoxicity is not due to polyphenols. If then what are other natural products present playing the role of cytotoxicity? However, major polyphenols are cytotoxic. There is a contrast of comment on line 154-157, with line 164-166. Justify your comment.

Authors’ response: We justify our comment

  1. Why the reversion factor increased from 32.95 to 362.50 fold? Seems very large change.

Authors’ response: We check our results and they are right, but include discussion about the key factors that affect the results.

  1. Better to add assay images of methanol extract to demonstrate potency.

Authors’ response: We include assay images.

  1. I recommend the GC-MS analysis of most potent methanol extract for the identification of bio-active components.

Authors’ response: We can´t make it now, because the project is finished but we are considering it in a second step.

Reviewer 2 Report

The manuscript investigated the In vitro study of the antiproliferative activity and modulation of cytotoxicity of C. chinense by-products extracts obtained from plants.  The subject frame of the work is well constructed. So, in this respect and this article should be contributed to present research. I recommended this work for publication after the following minor revisions.

1.      There are several typographical mistakes as well in whole manuscript. Therefore, the author’s thoroughly careful check the language and typo mistake to minimize the error.

2.      The abstract should be beginning with a sentence about the background of concept and the aims as well as novelty of study should be mentions. What exactly is the novelty of this study? The abstract is poorly written and should be improved. Abbreviations must be avoided in abstract. Parenthesis should be avoided in abstract - this is poor writing. Please improve.

3.      Introduction; Check and format the citations in the whole manuscript. Also, Appropriate references must be provided to explained the background, what is already done and why this study carried out. Other vise the novelty of this research is still poorly presented. This is important especially for the high IF journals. The scientific style should be used. What exactly is the aim of this work? Hypothesis statement is missing in the introduction section.

4.      Results and discussion; General remark to the discussion - In my opinion, the discussion provided by Authors is difficult to follow and verify due missing critical details in the methodology section. Due to poorly described material and poorly presented methods, I am not able to follow and properly review the discussion.

5.      All figures are of poor technical quality and not suitable for publication, especially in a high reputed journal. Font size and kind is too small and must be unified in all figures. Small writings are unreadable. All figures must be self-explanatory. Axis titles are poorly presented or absent. Units are missing. Are the data presented in figures significantly different? At least error bars should be shown.

6.      I suggest first time write full name rather than abbreviation; revise throughout in manuscript

Author Response

Reviewer 2

The manuscript investigated the In vitro study of the antiproliferative activity and modulation of cytotoxicity of C. chinense by-products extracts obtained from plants.  The subject frame of the work is well constructed. So, in this respect and this article should be contributed to present research. I recommended this work for publication after the following minor revisions.

  1. There are several typographical mistakes as well in whole manuscript. Therefore, the author’s thoroughly careful check the language and typo mistake to minimize the error.

Authors’ response:  We corrected the mistakes

  1. The abstract should be beginning with a sentence about the background of concept and the aims as well as novelty of study should be mentions. What exactly is the novelty of this study? The abstract is poorly written and should be improved. Abbreviations must be avoided in abstract. Parenthesis should be avoided in abstract - this is poor writing. Please improve.

Authors’ response: We corrected the abstract

  1. Introduction; Check and format the citations in the whole manuscript. Also, Appropriate references must be provided to explained the background, what is already done and why this study carried out. Other vise the novelty of this research is still poorly presented. This is important especially for the high IF journals. The scientific style should be used. What exactly is the aim of this work? Hypothesis statement is missing in the introduction section.

Authors’ response: We rewrite the introduction

  1. Results and discussion; General remark to the discussion - In my opinion, the discussion provided by Authors is difficult to follow and verify due missing critical details in the methodology section. Due to poorly described material and poorly presented methods, I am not able to follow and properly review the discussion.

Authors’ response: We describe a little more the methods

  1. All figures are of poor technical quality and not suitable for publication, especially in a high reputed journal. Font size and kind is too small and must be unified in all figures. Small writings are unreadable. All figures must be self-explanatory. Axis titles are poorly presented or absent. Units are missing. Are the data presented in figures significantly different? At least error bars should be shown.

Authors’ response: We changed the figures and we changed the figure 3 for a table.

  1. I suggest first time write full name rather than abbreviation; revise throughout in manuscript

  Authors’ response: We write full names

Reviewer 3 Report

The manuscript titled "In vitro study of the antiproliferative activity and modulation of cytotoxicity of C. chinense by-products extracts obtained from plants grown on black and red soils of Yucatán, México" is well-elaborated research that continues the work of the authors on Capsinum spp. by-products evaluation.

In this work, the extracts of leaves, pedunculus, and stems of C. chinense were obtained by three different methods (maceration, Soxflet, and supercritical fluid extraction), and then were evaluated by SRB assay, on cancer (HeLa, MCF-7, HCT-116, HCT-15, Caov-3), and normal cell line (184B5 breast normal cell line), respectively. Thereafter, the efficacy of cytotoxic modulation against MCF-7/Sens, MCF-7/Vin-, and MCF-7/ 203 Vin+ cell lines was determined.

In general, the data are strong and convincingly show the importance of studying diverse extracts from plant by-products obtained in a different types of soils to probably propose new therapeutic strategies against human diseases. The manuscript is well written and concise and the appropriate references are cited.

Overall, this is a well-performed study that I consider that is important and could be published after minor revision.

The authors need to address the below comments to strengthen the quality of the manuscript:

Include in the title the full name of the plant (Capsicum chinense).

Add the refernce/s for the statements presented in lines: 58-61.

Add the abbreviations used in Table 1.

Check the text for minor misspellings and/or rewrite the phrases from lines 167-169, 204-205, 221-223, 263-266.

Write all the chemical names with small caps inside the phrase (e.g. line 150).

Correct Figure 3 (y axis).

In the M&M add the concentrations of the extracts used in the analyses (e.g. mg L-1) and that of the cultured number of cells.

For the software add information on the city and country of origin.

Author Response

Reviewer 3

  1. The authors need to address the below comments to strengthen the quality of the manuscript:

Include in the title the full name of the plant (Capsicum chinense).

Authors’ response: We include in the title the full name of the plant

  1. Add the refernce/s for the statements presented in lines: 58-61.

Authors’ response: We changed this paragraph and including the references

  1. Add the abbreviations used in Table 1.

Authors’ response: We add the abbreviations

  1. Check the text for minor misspellings and/or rewrite the phrases from lines 167-169, 204-205, 221-223, 263-266.
    Authors’ response: We changed the errors

  1. Write all the chemical names with small caps inside the phrase (e.g. line 150).

Authors’ response: We write the chemical names with small caps inside the phase

  1. Correct Figure 3 (y axis).

Authors’ response: We changed the figure 3 for a table

  1. In the M&M add the concentrations of the extracts used in the analyses (e.g. mg L-1) and that of the cultured number of cells.

Authors’ response: We include in materials and methods the concentrarion of the extracts used in the analysis and that of the cultured number of cells

  1. For the software add information on the city and country of origin.

Authors’ response: The source data of the software has already been included

Round 2

Reviewer 1 Report

The manuscript can be accepted now.

Reviewer 2 Report

No further comments